# Electric control of spin transitions at the atomic scale

Piotr Kot[1], Maneesha Ismail[1], Robert Drost[1], Janis Siebrecht[1], Haonan Huang [1] & Christian R. Ast [1] ✉

Electric control of spins has been a longstanding goal in the field of solid state physics due to the potential for increased efficiency in information processing. This efficiency can be optimized by transferring spintronics to the atomic scale. We present electric control of spin resonance transitions in single TiH molecules by employing electron spin resonance scanning tunneling microscopy (ESR-STM). We find strong bias voltage dependent shifts in the ESR signal of about ten times its line width. We attribute this to the electric field in the tunnel junction, which induces a displacement of the spin system changing the *g*-factor and the effective magnetic field of the tip. We demonstrate direct electric control of the spin transitions in coupled TiH dimers. Our findings open up new avenues for fast coherent control of coupled spin systems and expands on the understanding of spin electric coupling.

Spintronics and the concept to control spin and magnetic properties using electric fields have been on the forefront of solid-state research for the past several decades with the promise to increase efficiency in data processing[1–4]. Different concepts have been considered, such as the spin transistor[5–8], the spin Hall effect[9,10], dopants in silicon[11–13], and magnetic molecules[14–22]. Specifically, spin-electric control allows for superior scalability and switching as electric fields are more easily contained and faster to manipulate than magnetic fields. This type of processing could be further optimized by transfering it to the atomic scale, for which scanning tunneling microscopy (STM) is an ideal platform in realizing such a goal. Specifically, the combination of electron spin resonance spectroscopy (ESR) with STM has expanded the sensitivity of ESR to atomic scale spin systems, and has enhanced the attainable energy resolution of STM well into the neV range[23–27].

As the manipulation capabilities in STM are mostly based on electrical control, implementing sizeable atomic-scale electrical spin control can become not only possible with ESR–STM, but also quite effective. The applied bias voltage typically induces a very strong electric field between the tip and sample due to the extremely small gap of only a few Ångströms[28]. Moreover, ESR spectra are typically acquired by sweeping the microwave frequency or the magnetic field, so that the bias voltage essentially becomes a free parameter to be tuned. Also, electric control provides a degree of freedom that can be

manipulated on a fast timescale, which is a promising avenue to coherent control of atomic spin states[29]. However, so far the DC bias voltage in ESR-STM has not been employed for spin manipulation.

In this study, we exploit the bias voltage as an electrical means for direct manipulation of spin transitions. We use a TiH molecule on an insulating MgO layer (see Fig. 1a) to demonstrate a direct tuning of the *g*-factor and the tip magnetic field. In this system, the resonance peak shifts by many line widths within a bias voltage range of 240 mV (see Fig. 1b), which is much stronger than what has been predicted for this system (on a different adsorption site)[30] or previously measured in bulk systems[15]. This effect can be seen in Fig. 1c for individual magnetic field sweeps of the ESR peak at different bias voltages. The ESR peaks are well separated from each other. We explain this effect by the strong electric field in the tunnel junction induced by the applied bias voltage and felt by the dipolar TiH molecule. A change in the electric force shifts the equilibrium position of the TiH molecule, resulting in the *g*-factor being modified and the molecule feeling a different magnetic field from the spin-polarized tip. The *g*-factor is, in part, modified due to a change in the crystal field felt by the TiH[30]. We estimate this electric field-induced displacement to be on the order of −11.5 fm/mV, which amounts to a total shift of about 2.7 pm, i.e., about 1% of the estimated equilibrium distance of TiH over MgO[30,31] (see Supplementary Note 1 for details).

[1]Max-Planck-Institut für Festkörperforschung, Heisenbergstraße 1, 70569 Stuttgart, Germany. ✉e-mail: c.ast@fkf.mpg.de

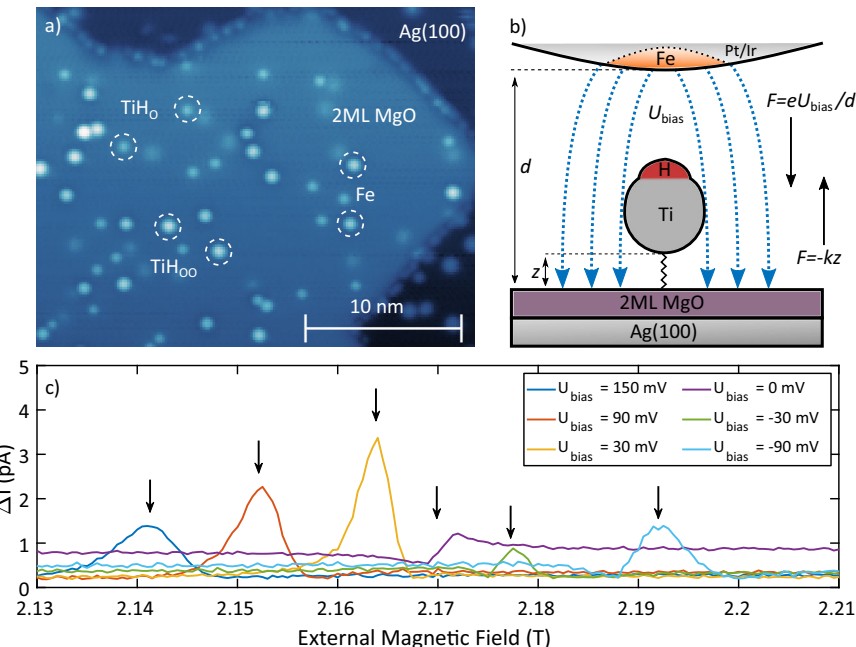

**Fig. 1 | ESR on TiH molecules. a** STM topography of 2 ML MgO on a Ag (100) substrate decorated with individual TiH molecules and Fe atoms ($U_{sp} = 100$ mV, $I_{sp} = 20$ pA). The different species are labeled and circled accordingly. **b** Schematic of the tunnel junction during the ESR experiment. Force vectors representing the electric force induced by the bias voltage and elastic force of the Ti-MgO bond are shown. Additionally, the electric forces may act on the Ti-H bond. **c** Magnetic field sweeps performed at different STM junction bias voltages $U_{bias}$ ($U_{sp} = 100$ mV, $I_{sp} = 250$ pA, $f_{rf} = 61.545$ GHz, $U_{rf} = 20$ mV). The ESR peak positions are labeled with black arrows. The spectra are constant voltage slices of the data in Fig. 2b.

## Results and discussion

The measurements were done on TiH molecules that adsorb on the bridge-site between two O atoms of the MgO double layer at a base temperature of 310 mK and in a magnetic field that is oriented perpendicular to the sample surface. The molecules are labeled as $TiH_{OO}$ in Fig. 1a. Varying the bias voltage continuously, we observe the evolution of the ESR peak as a function of both bias voltage and external magnetic field at a constant microwave radiation frequency of 61.545 GHz and a microwave amplitude of 20 mV. This is shown for two different setpoint currents of $I_{sp} = 100$ pA and $I_{sp} = 250$ pA in Fig. 2a and b, respectively. Unless otherwise noted, the corresponding setpoint voltage is $U_{sp} = 100$ mV. Several ESR-sensitive tips were used during the course of this study with all of them showing substantial spin-electric coupling (SEC). The data presented in this manuscript was measured with two of these microtips. For more information please refer to Supplementary Note 2. The horizontal features in Fig. 2a, b are due to the radio frequency-induced rectification of the nonlinear $I(V)$ response of the junction and are not related to the ESR signal[32-35]. Comparing the slope of the ESR peak in Fig. 2a, b, we directly see that the change in the resonance condition is more pronounced for the higher setpoint current, which points towards an influence of the electric field rather than the bias voltage. We have obtained similar results for TiH molecules adsorbed on top of an O atom of the MgO layer (labeled $TiH_O$ in Fig. 1a), which are presented in Supplementary Note 3.

For a more quantitative analysis of the evolution of the ESR peak, we exploit the linear dependence of the ESR resonance on the magnetic field as

$$E_Z = hf_{res} = g\mu_B(B_{ext} + B_{tip}), \quad (1)$$

where $E_Z$ is the Zeeman energy, $f_{res}$ is the resonance frequency, $g$ is the $g$-factor, and $B_{ext,tip}$ are the external magnetic field and the field of the tip felt by the spin system (henceforth the tip field), respectively. Furthermore, we assume the spin to be $S = 1/2$[36], $h$ is Planck's constant,

and $\mu_B$ is the Bohr magneton. Both the tip field $B_{tip}$ and the $g$-factor will be a function of the electric field in the junction, which in turn is a function of the applied bias voltage. We note that the tip field may depend on the $g$-factor as has been discussed previously[37,38]. Analyzing the data at different frequencies, we extract the $g$-factor and the tip field $B_{tip}$ dependency on the bias voltage at four different setpoint currents, which is shown in Fig. 2c, d (for details on the curve fitting, see Supplementary Note 4). We can clearly see that both the $g$-factor and the tip field $B_{tip}$ monotonically increase with increasing bias voltage. This indicates that both quantities are sensitive to the changing electric field. In addition, the change is stronger at a larger setpoint current, which is consistent with our interpretation as a smaller tip-sample distance will lead to a stronger adjustment of the electric field with respect to bias voltage.

One notable difference in the behavior of the $g$-factor and the tip field $B_{tip}$ is around zero bias voltage, where the effects of the electric field vanish. Interestingly, near-zero bias voltage the tip field is relatively stagnant as a function of the set point current, while the increase in the $g$-factor is comparable to non-zero bias voltages. Calculations in the literature show that the $g$-factor increases as the molecule-substrate distance decreases for $TiH_O$[30,31] (we expect a similar behavior for $TiH_{OO}$). We have measured approach curves demonstrating that the molecule-substrate coupling increases as the tip-sample distance is reduced. This indicates a decrease in the molecule-substrate distance, which provides an overall consistent behavior for the increasing $g$-factor for larger set point currents (see Supplementary Note 5 for details). Our findings show that adjusting the tip-sample distance results in changes to both the tip field and the $g$-factor. The changes due to the tip-sample distance have previously been attributed to the tip field[36,39,40], while theoretical considerations of an electric field dependence have not taken changes in the tip field into account[30]. However, as we show here, the two effects cannot be easily separated. For a more detailed understanding of the tip field influence, the contributions from dipolar and exchange interactions[40] as well as from atomic, elastic, and electric forces have to be combined as they

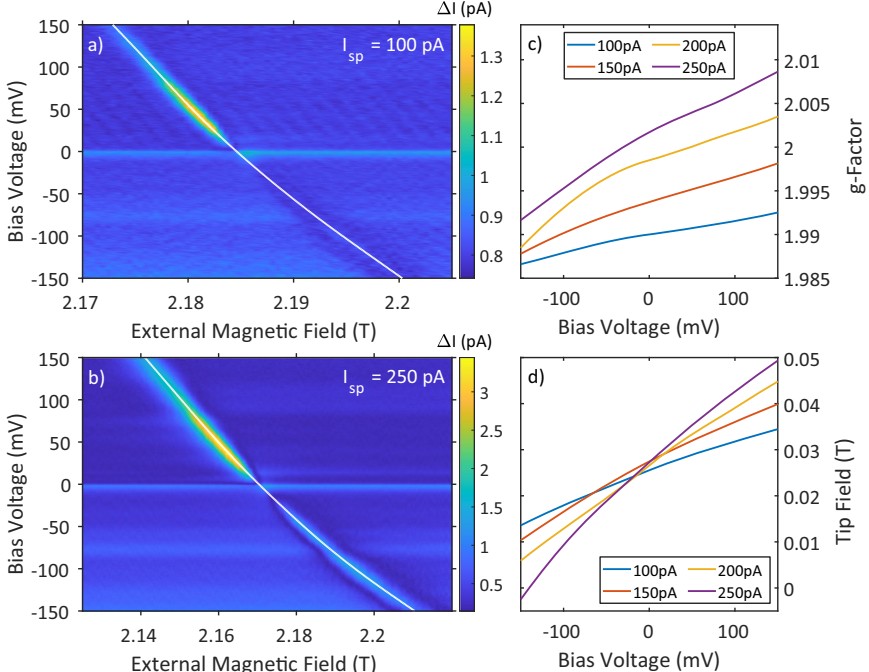

**Fig. 2 | Voltage dependence of the ESR signal.** Magnetic field/bias voltage sweeps performed at two different current set points ($U_{sp}$ = 100 mV, $f_{rf}$ = 61.545 GHz, $U_{rf}$ = 20 mV, **a** $I_{sp}$ = 100 pA, **b** $I_{sp}$ = 250 pA). White dashed lines show a spline interpolation to the ESR peak positions as a function of bias voltage. **c**, **d** Extracted g-factor and tip field vs. bias voltage at four current set points.

seem to have at least partially opposing effects (cf. Fig. 2d). Since the g-factor only changes by about 1% in the measured parameter space, we expect the effects regarding the aforementioned dependence of the tip field $B_{tip}$ on the g-factor to be small compared to its dependence on the tip-sample distance. Interestingly, at $I_{sp}$ = 250 pA near $U_{sp}$ = − 140 mV, the tip field vanishes, which could be used as a "no tip influence point" reducing tip-related errors and uncertainties in the analysis as discussed previously[40]. We note that the dependence of the g-factor on the tip-sample distance can explain the smaller tip fields $B_{tip}$, which we find compared to previous reports[36,39,40].

To compare our results with literature, we calculate an effective frequency shift as a function of applied bias voltage of 0.83 GHz/V and 4.3 GHz/V for the g-factor and the tip field, respectively, at a setpoint current of 250 pA (see Supplementary Note 4 for details). These values are orders of magnitude larger than what has recently been reported for the ESR peak shift of 5.7 kHz/V in a bulk matrix of $HoW_{10}$ nanomagnets[15]. We can reach these values because the electric field becomes extremely large between the tip and sample. Comparing the SEC constants, which relate the frequency shift to the applied electric field, the situation looks a bit different. For the $HoW_{10}$ nanomagnets[15], a value of 11.4 Hz/(V/m) was reported, while we estimate values of 0.4 Hz/(V/m) and 2.2 Hz/(V/m) for the g-factor and the tip field $B_{tip}$, respectively, assuming a tip-sample distance of about 5 Å. While this indicates a more efficient coupling mechanism for the $HoW_{10}$ nanomagnets, the particular TiH system was not optimized a priori for high SEC, so we anticipate spin systems with superior SEC to be identified in the future.

Furthermore, the response of the Zeeman splitting to an electric field has been previously calculated specifically for the TiH molecule on MgO, albeit on an oxygen site TiH rather than on bridge site TiH[30]. The calculated frequency shift is estimated to be about 0.2 GHz/V, which is smaller than what we have observed experimentally. Neglecting the effect of the tip field, which was not considered in the calculations, we find a four times stronger change in the frequency shift for the g-factor in the experiment. We surmise that additional changes other than the crystal field gradient and the equilibrium

position of the whole TiH molecule, such as a change in the Ti-H bond or simply the different adsorption site, contribute to this difference. The sensitivity of the TiH molecule to the local environment is already illustrated by changing the spin state from 3/2 in the gas phase to 1/2 upon adsorption on the surface, as well as changing the g-factor from about 2 to 0.6 by moving to a different binding site on the MgO[31,41]. The ability to tune the g-factor and the tip field $B_{tip}$ by means of the bias voltage demonstrates a degree of freedom for in situ electrical manipulation of the spin transitions.

We demonstrate direct manipulation through SEC on two different types of dimers with different distances between the TiH molecules[36,42]. In the first example, the two bridge site TiH molecules ($TiH_{OO}$) are 644 pm apart (see inset in Fig. 3a), such that the coupling is relatively strong ($J \approx 61.1$ GHz). We identify three transitions in this dimer in Fig. 3a. These transitions (labeled I, II, and III) are well separated near zero bias voltage and subsequently broaden as well as intersect as we increase the bias voltage[42]. The white dashed lines are calculated ESR peak positions from a Hamiltonian model for coupled spins assuming a linear dependence of the g-factors and the tip field $B_{tip}$ on the bias voltage (for details on the model and the parameters see Supplementary Note 6). The corresponding energy levels at a constant external field of 2.2 T are plotted in Fig. 3b with the transitions being indicated. We identify transition III as a clock transition that would not be visible if the two g-factors in the dimer were equal[42]. Therefore, we know that the two g factors are not equal even at zero bias voltage. Furthermore, as shown in Fig. 3a we can tune transitions II and III such that they are located at the same external magnetic field value, which demonstrates that we can manipulate the spin transitions in a dimer by means of SEC.

If the two TiH molecules are 1.04 nm apart (see inset in Fig. 3d), the interaction between them is reduced ($J \approx 0.67$ GHz), which shifts the energy of the singlet state $|S\rangle$ close to the triplet state $|T_0\rangle$ as shown in Fig. 3c[42,43]. The singlet state $|S\rangle$ and the triplet state $|T_0\rangle$ undergo an avoided crossing (see inset in Fig. 3c), which can be observed experimentally[43]. We have tuned the tip-sample distance such that we can observe this avoided crossing in a bias voltage range

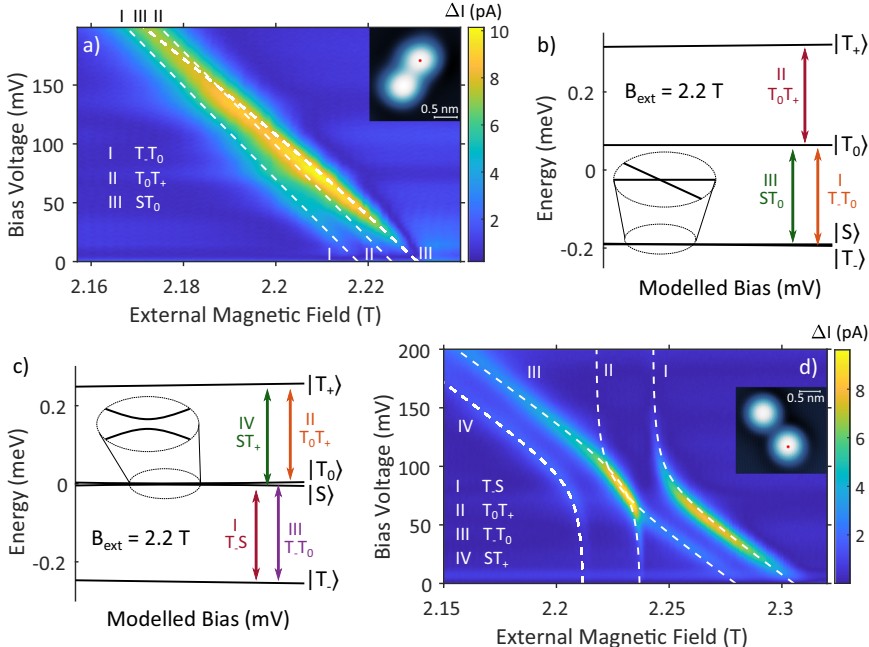

**Fig. 3 | Interaction tuning in dimers. a** Magnetic field/bias voltage sweep on a strongly coupled dimer ($U_{sp} = 150$ mV, $I_{sp} = 1$ nA, $f_{rf} = 61.545$ GHz, $U_{rf} = 20$ mV). The inset shows the topography of the dimer with the red dot indicating the position of the tip during measurements ($U_{sp} = 100$ mV, $I_{sp} = 20$ pA). **b** Modeled behavior of the spin states at a constant external magnetic field that results in the ESR transitions measured by the experiment in (**a**). The colored arrows show the observed transitions. **c** Modeled behavior of the spin states at a constant external magnetic field that result in the ESR transitions measured by the experiment in (**d**). The colored arrows show the observed transitions. **d** Magnetic field/bias voltage sweep showing the avoided crossing of two coupled TiH molecules ($U_{sp} = 100$ mV, $I_{sp} = 400$ pA, $f_{rf} = 61.545$ GHz, $U_{rf} = 20$ mV). The inset shows the topography of the dimer with the red dot indicating the position of the tip during measurements ($U_{sp} = 100$ mV, $I_{sp} = 20$ pA). The white dashed lines in (**a**) and (**d**) are calculated ESR peaks from a Hamiltonian model for coupled spins (see Supplementary Note 6 for details).

between 0 mV and 200 mV as shown in Fig. 3d. The four transitions that are visible in Fig. 3d are labeled I through IV corresponding to the transitions indicated in Fig. 3c. We can clearly see how the two pairs of transitions associated with each TiH molecule in the dimer approach the avoided crossing and separate again. The white dashed lines are calculated ESR peak positions using the same dimer spin Hamiltonian as before, just with a weaker exchange interaction, which corroborates the experimental observations (for details on the parameters, see Supplementary Note 6). For smaller magnetic fields below the avoided crossing, the transitions III and IV are strongly influenced by the SEC, which indicates that the wave functions of the corresponding energy levels are located on the TiH molecule under the tip. As transitions I and II are much less influenced by the applied bias voltage, we conclude that the corresponding wave functions are located on the TiH molecule next to the tip. The slope is not vertical, so we expect some influence of the electric field on the TiH molecule next to the tip about 1 nm away. For higher magnetic fields above the avoided crossing, the situation is reversed, such that the wave functions for transitions I and II are in the TiH below the tip and the wave functions for transitions III and IV are in the TiH next to the tip.

The ability to manipulate spin interactions in dimers through SEC clearly demonstrates the versatility of voltage-dependent ESR-STM. However, the tunneling current itself is the biggest source of decoherence in the ESR excitation[44]. As a final proof-of-principle, we exploit both the bias voltage and the tip-sample distance as two degrees of freedom to move the avoided crossing to zero bias voltage, where the tunneling current is minimized and correspondingly the coherence time is maximized. This will, for example, allow for tuning in and out of the coherent evolution of entangled states in a TiH dimer using the bias voltage instead of the tip-sample distance as has recently been demonstrated[43].

In order to move the avoided crossing of the second TiH dimer in Fig. 3d to zero bias voltage, we increase the tip-sample distance such

that the setpoint reduces from $U_{sp} = 100$ mV and $I_{sp} = 400$ pA to $U_{sp} = 50$ mV and $I_{sp} = 112$ pA. Here, the avoided crossing shifts in bias voltage when adjusting the tip-sample distance, but essentially remains at the same position in external magnetic field. Figure 4a shows the corresponding measurement, where the avoided crossing is now moved close to zero bias voltage. At zero bias voltage, only the homodyne detection scheme allows to observe the ESR peaks, which typically appear as asymmetric peaks[42]. This can be seen in Fig. 4b for three different current setpoints, where the avoided crossing is above zero voltage (blue), near zero voltage (red), and below zero voltage (yellow). The shifts of the resonances corresponding to the movement of the avoided crossing in bias voltage is clearly visible. This demonstrates that by considering the bias voltage in ESR-STM, we can manipulate spin structures in a more complex manner than previously possible.

The ability to tune spin transitions at the nanoscale by means of an electric field opens up many interesting possibilities in the atomic manipulation capabilities of STM far beyond the proof-of-principle presented here. It adds the otherwise unconsidered bias voltage to the degrees of freedom for customizing spin systems to specific needs. In this regard, the tip-sample distance, which has previously been used, and the bias voltage present ideal tuning parameters for manipulating complex spin structures. Furthermore, the bias voltage opens avenues towards a more complete understanding of the ESR mechanism in the STM and its dynamics as well as its sources of decoherence and dissipation. This becomes particularly interesting for future applications in time-resolved experiments as it enables fast switching schemes for the bias voltage, which is not possible for magnetic fields or the tip-sample distance (e.g., coherent evolution[43], qubit operations[45,46]). Specifically, our findings present an important step in implementing coherent control in spin states that would lead to atomic-scale quantum information processing. We believe that all atoms/molecules will be more or less susceptible to an electric field so that they will show

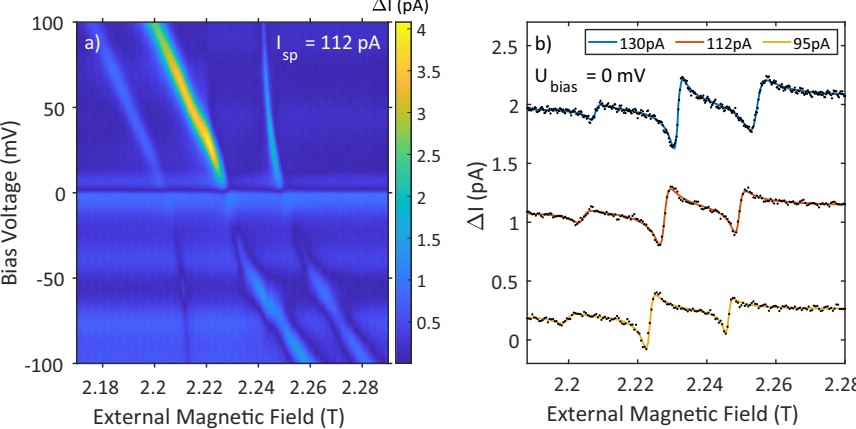

**Fig. 4 | Tuning the avoided crossing. a** Magnetic field/bias voltage sweep showing the avoided crossing of the TiH dimer in Fig. 3c near zero bias voltage ($U_{sp}$ = 50 mV, $I_{sp}$ = 112 pA, $f_{rf}$ = 61.545 GHz, $U_{rf}$ = 20 mV). **b** ESR sweeps measured at zero bias showing how the resonances shift with respect to current set point ($U_{sp}$ = 50 mV, $f_{rf}$ = 61.545 GHz, $U_{rf}$ = 20 mV). This shows that the tip-sample distance can be adjusted to bring the avoided crossing exactly to zero bias.

some kind of SEC, which makes it a general phenomenon to be considered[47]. Moreover, our study opens up new directions by which quantities could be electrically controlled in an STM junction. For example, the electric field dependence of magnetic anistropies could play a role for higher-order spins, which presents an interesting path for future work. Looking on a broader perspective, we have established SEC in ESR-STM, which connects to the well established field of spintronics on an atomic scale. Moreover, studying the influence of the electric field within ESR-STM opens possibilities for a better understanding of optimizing SEC in bulk materials.

Indeed, we believe that the crystal field is affected by these strong local electric fields, which leads to a change in the *g*-factor as discussed in ref. 30. There may certainly be more quantities that are affected by the electric field, which provides exciting perspectives for future research opening up a number of new possibilities for spin manipulation. Anisotropy may also play a role for higher-order spins, which likely are just as susceptible to the electric field. This also presents interesting directions for future work. We, unfortunately have not been able to test this procedure on Fe atoms due to experimental limitations in our setup.

## Methods

Experiments were performed using a commercial Unisoku USM-1300 STM retrofitted with high-frequency cabling and antenna. The DC bias voltage was applied on the sample and the current was measured from the tip for all measurements except for the measurements on oxygen-site TiH molecules (Supplementary Note 5 only), where the connections were interchanged. The high-frequency setup allows for driving ESR signals between 60 GHz to 100 GHz[27]. Similarly to what has been shown previously[24], we calibrate $U_{rf}$ in our junction by measuring the radio frequency reponse of nonlinearities in tunneling spectroscopy[27]. We cleaned Ag(100) in UHV by repeated cycles of Ar$^+$ ion sputtering at 5 kV and annealing at 820 K. MgO was grown on the clean Ag by simultaneous evaporation of Mg onto the sample surface, leaking of $O_2$ into the UHV space, and heating of the Ag substrate. Deposition times varied from 15 to 20 min with the Mg Knudsen cell being heated to 500 K, the $O_2$ being leaked to $10^{-6}$ mbar and heating of the Ag to 520 K. After the MgO growth, we deposited Fe and Ti on the surface using e-beam evaporators by applying an emission voltage of 850 V and an emission current of 8.5 mA for Fe and 19 mA for Ti. Furthermore, the sample was kept below 16 K during Fe and Ti deposition to ensure that the atomic species did not form clusters on the surface. The Ti species naturally hydrate due to the residual hydrogen gas found in the UHV space[36]. To create ESR-sensitive tips, we picked up between one and

ten Fe atoms[48]. Dimers studied in this letter were either found naturally occurring on the sample or were created via atom manipulation[49].

## Data availability

All data needed to evaluate the conclusions are present in this paper and/or the Supplementary Information. In addition, the data related to this paper are available from the EDMOND Database[50].

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

## Acknowledgements

The authors would like to thank Juan Carlos Cuevas, Andreas Heinrich, Klaus Kern, Jose Lado, Sander Otte, and Aparajita Singha for fruitful discussions. We are grateful to the European Research Council (ERC) for their financial support. This study was partly funded by the ERC Consolidator Grant AbsoluteSpin (Grant No. 681164) and by the Center for Integrated Quantum Science and Technology (IQ$^{ST}$).

## Author contributions

P.K. and M.I. carried out the measurements with the help of J.S. and R.D. P.K., H.H. and C.A. analyzed the data. All authors interpreted the results. P.K. and C.A. wrote the manuscript. C.A. supervised the project.

## Funding

## Competing interests

The authors declare no competing interests.
