## [Peer Review File · Nature Communications]

Reviewers' Comments:

Reviewer #1:

Remarks to the Author:

In the manuscript titled "Electric Control of Spin Transitions at the Atomic Scale" submitted by Kot et al., the authors perform STM-ESR experiments at 310 mK on TiH molecules adsorbed on the bridge (and oxygen) site of a double layer of MgO/Ag(100). By utilizing the bias voltage, a new tuning parameter, alongside the external magnetic field or the microwave frequency, is introduced to tune the spin state of the investigated molecule or dimer. This procedure offers an all-electrical ultrafast method of manipulation, which can be helpful for future experiments for the coherent control of spin states.

Overall, the manuscript is very well written and structured, the figures are presented clearly and the objective of the experimental investigations is intelligible. Furthermore, the experiments are carried out carefully and the methods are sound. In my opinion, the work is a natural step forward in the growing field of ESR-STM that will be useful for the community.

However, I have a few remarks that should be addressed prior publication in Nature Communications. Please see below:

1. While the instrumental characteristics are not the focus of this manuscript, the authors should more clearly refer to their own work in order to make it a bit clearer what the high-frequency characteristics of the setup are and how, e.g., the V_{rf} has been calibrated. My suggestion is to add a few sentences about this by citing ref. [27] (Drost et al., RSI 93, 043705 (2022)). This could be done in the methods section.
2. Just a small question about the Ag preparation: Was the Ag really sputtered at 5 kV? It seems quite high to me.
3. In Fig. 2 a, b, the authors show spline fits to the ESR peak positions as a function of bias voltage. For me, a spline is not a fit but an interpolation method. Hence, it is not a fit in a physically meaningful sense. The authors may correct me if I am wrong. But if not, then I think the wording here is misleading and I would change the wording from "fit" to "interpolation".
 - a. Furthermore, Fig. 3 then shows "fits to the ESR peak positions". For me it is not clear if this is again a spline fit or an actual fit of the ESR peaks. Please refine the wording to be more clear. If these are proper fits now, then I suggest to change the dashed lines to dotted lines in Fig. 2, to distinguish both better.
 - b. Depending on the above, the wording in and around Fig. S3 should then also be adjusted.
4. Compared to other ESR-STM publications on this sample system, the setpoint parameters, especially the current I_{sp} with up to 1 nA, are quite large. Derived from this, I have some questions:
 - a. Do the authors know the influence of the tip on their spin states?
 - b. Can a tip stray field be estimated from their measurements? They talk about the tip field in the supplement and find very small tip fields of only a few mT from their fits. These are extremely small numbers especially at these setpoints. For comparison, in PRL 119, 227206 (2017) tip fields of up to hundred mT are estimated, with much smaller setpoint parameters. Can the authors comment on this? I wonder if the influence of the tip field might be stronger than anticipated here.
 - c. Are there differences in lineshapes of the ESR peaks the authors observe at different current setpoints or tip-sample distances?
5. A general question about the mechanism of the spin-electric coupling: Is it a possibility that not also other degrees of freedom besides the spin are affected by the applied biases and can lead to similar effects? E.g. the crystal field might be affected by the strong local electric fields since the MgO is heavily ionized. This would change the whole energy landscape of the adsorbed molecule. What role does anisotropy play here? And along these lines, is this method even viable for spins >

½? Have the authors tried the same procedure on the Fe atoms that are co-deposited?

6. In terms of the TiH molecule, it has been shown that the Oxygen-site molecules exhibit i) an orbital excitation at around 90 meV (ref. [31]) and ii) anharmonic excitations of the dynamic quantum system consisting of hydrogen and Ti atom starting from ca. 35 meV. How can it be excluded that at biases above these thresholds such excitations do not play a role in the interpretation of the data?

Reviewer #2:

Remarks to the Author:

This manuscript explores the potential for controlling spin transitions through the application of a constant electric field generated between the tip of an STM and the surface atom. The authors devote half of their research to investigating spin control in an isolated TiH, while the remaining half is focused on the study of two different TiH dimers. The study is well presented and represents a significant contribution to this field. The ability to control spins electrically using these techniques has been a subject of interest for a relatively brief period (last two years may be). This study is the first to clearly demonstrate (experimentally) the feasibility of such control.

Overall, the authors' work is an important contribution to the field and sets a new standard for future research.

Despite what was said above, I believe that the authors need to improve the theoretical description of the observations made in this experiment. Specifically, it appears that Equation (1) in the manuscript is incomplete and/or may be lacking some essential information that is required to fully understand its meaning and implications. The total field experienced by the TiH arises from the external magnetic field and the field generated by the tip, as the authors point in this work. However, in the paper by Willke et al. (Willke, et al "Magnetic resonance imaging of single atoms on a surface" Nature Physics, 15 (10), (2019)), it is shown that the field created by the STM tip has two contributions: the dipolar and exchange components. While the present study briefly mention these contributions, they are not discussed in detail. Willke et al. clearly demonstrate that both contributions are substantially different (see Eq about dipolar field and discussion below), particularly regarding the exchange's effect on the resonant frequency, which is independent of g (the effect of the exchange field on the resonance frequency does not depend on g). It is unclear how this is reconciled with equation (1) in the current study. In order to write the resonance frequency as in Eq. (1) we need that the tip's field depends on g . This creates some ambiguity and potentially contaminates the interpretation of Fig. 2 of the present manuscript (c and d). Therefore, the authors should improve the discussion of Eq (1), discussion of Fig. 2, and above all, discussion of the dependence of the field created by the tip on the electric field because, as Eq. (1) is presented, we would have an implicit dependence of the magnetic field on the g -factor. After clarifying the issue with equation (1), I believe it would be interesting to have a more detailed discussion about the different mechanisms involved in the electrical control. It is known that the electric field of the tip modulates both the g -factor and the magnetic field of the tip (as shown in figures 2c and 2d). Have the authors been able to place the STM tip in positions where its field (of the tip) is not present (as shown in the reference [39] of this paper and known as notinfluence point - NOTIN)? This would allow for control to be achieved mainly through the g -factor. It would be helpful if the authors could provide some discussion about this issue.

I think that the second half of the manuscript is quite nice. The discussion about control in dimmers is, from my point of view, the most interesting part of the work. The figure 3-d is a good

one and, I think it gives us a complete idea of what is possible to do in this kind of experiments.

The conclusions of the work are also clear and insightful, making the reading experience quite enjoyable.

I highly recommend that the authors address the issue highlighted at the beginning of this report. Once this concern has been effectively addressed, I believe that this work should be considered for publication in Nature Communications.

Reviewer #3:

Remarks to the Author:

The manuscript describes a detailed experimental investigation of the control of TiH spin transitions indicating that the resonance condition can be tuned by crystal field modulation of the g factor, as well as the STM tip's local field. These dependencies are then leveraged and applications are demonstrated in coupled quantum systems. The paper advances the quantitative understanding of spin systems in STM and importantly the techniques shown as proof of principle here can be used in future investigations of atomic scale quantum systems. The authors provide an excellent level of detail on their experimental setup and methodologies and the quality of the data and presentation are very good. I recommend the paper for publication in Nature Communications after addressing some minor concerns.

1. Abstracts should contain important details such as the experimental system under study (TiH) and key results. The only thing that is quantitative in it is the statement "about ten times its line width". The current abstract is too generic. Control of g and local tip field should be mentioned, as well as applications to tuning interactions of coupled quantum systems. Quantitative results are encouraged.

2. "However, so far the bias voltage in ESR-STM has not been employed for spin manipulation" This is confusing because RF modulation of the bias voltage is what drives spin transitions in ESR-STM. I think the authors mean the average (DC) value of the bias voltage here - could they find a way to clarify what they mean?

3. Hopefully I didn't miss it, but please state somewhere in the paper or supplementary if U_{sp} refers to voltage applied to tip or sample for clarity.

4. I don't agree with the use of "ultrafast" in the abstract and second paragraph (no fs time scales).

5. "The horizontal features in Fig 2(a) and (b)" -- have the authors considered these horizontal background features could be due to rectification of the sinusoidal 20mV RF bias through the nonlinear $I(V)$ curve of the junction? That could be checked by repeating the DC bias voltage sweep while superimposing a non-microwave frequency AC modulation (e.g. 100kHz), or by numerically convolving the $I(V)$ curve with an arcsine distribution and subtracting the non-convolved $I(V)$. The authors' explanation is more exotic, but I'd like to make sure there is evidence for it.

6. I had difficulty following the fitting procedure on my first readings of the manuscript. I'm pretty sure I understand it now - it's not very complicated, but the sweeps over multiple parameters made it hard to follow what was going on in my first readings. Here are some suggestions to make it easier to follow:

6a. Supp. page 2: "We then use the spline fits at four different microwave frequencies and perform a linear fit at each bias voltage."

I would suggest detailing the procedure a bit more here. I think the spline is mostly a means of tracking the resonance peak position in the U_{bias} , B_{ext} space and to bridge the gap across ± 20 mV.

"We then use the spline fits at four different microwave frequencies as a means to query the peak position for an array of bias voltages (including those that pass through 0V). At each bias voltage, the linear Zeeman energy is fit to four points and yield the g-factor..."

I would suggest the authors perhaps add discrete markers on the U_bias +/- 150 fits showing the four points the fit line passes through.

6b. I am curious about some of the details of the spline fit. Please consider including some as you see fit. Does the spline fit pass through every maximum peak position in the U_bias, B_ext heat maps? Does it offer any de-noising or smoothing in the U_bias, B_ext space (does it have less degrees of freedom than the number of U_bias B_ext values that were swept?). How were the peak positions determined as input to the spline fit (local fitting? max(signal) point?). It seems like the spline fit is a tool to query the resonance condition at any U_bias, enabling a series of fits across U_bias and in particular through zero. Is that how I should think of it?

6c. Figure S3 (a) should have a label or caption indicating $f = 61.545$ GHz. Fig S3(b) should have a label or caption indicating $I_{sp} = 250$ pA.

6d. Supp. page 2: "along with the ESR resonance positions calculated from the fitted values". It's not clear if this refers to the spline fits or the linear fits, recommend to be more explicit : "calculated from the g and B_tip fit values for each U_bias, I_sp condition"

6e. Figure S4: Y axis label should be "External Magnetic Field" rather than "ESR position" -- ESR position sounds like it might be the Zeeman energy which it isn't, and the term External Magnetic Field is used consistently elsewhere in the paper.

6f. Supp page 2 "For the frequency dependence of the g-factor" is confusing (makes it sound like g factor is frequency dependent). I think the authors mean "For the frequency shift due to the g-factor bias dependence"

Reply to Referee's Comments

Changes in the manuscript are marked in blue.

Answers to Referee #1:

In the manuscript titled "Electric Control of Spin Transitions at the Atomic Scale" submitted by Kot et al., the authors perform STM-ESR experiments at 310 mK on TiH molecules adsorbed on the bridge (and oxygen) site of a double layer of MgO/Ag(100). By utilizing the bias voltage, a new tuning parameter, alongside the external magnetic field or the microwave frequency, is introduced to tune the spin state of the investigated molecule or dimer. This procedure offers an all-electrical ultrafast method of manipulation, which can be helpful for future experiments for the coherent control of spin states.

Overall, the manuscript is very well written and structured, the figures are presented clearly and the objective of the experimental investigations is intelligible. Furthermore, the experiments are carried out carefully and the methods are sound. In my opinion, the work is a natural step forward in the growing field of ESR-STM that will be useful for the community.

However, I have a few remarks that should be addressed prior publication in Nature Communications. Please see below:

We thank the referee for their positive review of our work.

1. While the instrumental characteristics are not the focus of this manuscript, the authors should more clearly refer to their own work in order to make it a bit clearer what the high-frequency characteristics of the setup are and how, e.g., the V_{rf} has been calibrated. My suggestion is to add a few sentences about this by citing ref. [27] (Drost et al., RSI 93, 043705 (2022)). This could be done in the methods section.

We have added two sentences in the methods sections specifying our high frequency range and how we calibrate U_{rf} in our junction. We thank the referee for this suggestion.

2. Just a small question about the Ag preparation: Was the Ag really sputtered at 5 kV? It seems quite high to me.

Yes, we indeed did use such a high parameter. We found that at this strong parameter we could more effectively remove previous MgO growths on the sample.

3. In Fig. 2 a, b, the authors show spline fits to the ESR peak positions as a function of bias voltage. For me, a spline is not a fit but an interpolation method. Hence, it is not a fit in a physically meaningful sense. The authors may correct me if I am wrong. But if not, then I think the wording here is misleading and I would change the wording from “fit” to “interpolation”.

a. Furthermore, Fig. 3 then shows “fits to the ESR peak positions”. For me it is not clear if this is again a spline fit or an actual fit of the ESR peaks. Please refine the wording to be more clear. If these are proper fits now, then I suggest to change the dashed lines to dotted lines in Fig. 2, to distinguish both better.

b. Depending on the above, the wording in and around Fig. S3 should then also be adjusted.

We thank the referee for pointing out this oversight. For clarification we have changed all instances of “spline fits” in our manuscript to be “spline interpolations”.

a. In Fig. 3 a) and d), the white dashed lines are actually fitted ESR signal positions that result from the Hamiltonian models of coupled spins discussed in the Supplementary Information. We have adjusted the caption of Fig. 3 and the main text discussing Fig. 3 to make this more clear.

b. We have adjusted the wording in Fig. S3 accordingly.

4. Compared to other ESR-STM publications on this sample system, the setpoint parameters, especially the current I_{sp} with up to 1 nA, are quite large. Derived from this, I have some questions:

a. Do the authors know the influence of the tip on their spin states?

b. Can a tip stray field be estimated from their measurements? They talk about the tip field in the supplement and find very small tip fields of only a few mT from their fits. These are extremely small numbers especially at these setpoints. For comparison, in PRL 119, 227206 (2017) tip fields of up to hundred mT are estimated, with much smaller setpoint parameters. Can the authors comment on this? I wonder if the influence of the tip field might be stronger than anticipated here.

c. Are there differences in lineshapes of the ESR peaks the authors observe at different current setpoints or tip-sample distances?

a. As in most ESR-STM studies, the influence of the tip on the spin state is combined in an effective tip field, which is typically found in the offset of a linear fit to the Zeeman splitting. In this sense, we do know the influence of the tip field on the spin states as plotted in Fig. 2(d).

b. Here, we would like to point out that a dependence of the g -factor on the tip-sample distance and the applied bias voltage has not been considered previously, such that changes in the Zeeman splitting have been effectively attributed to the tip field. Our analysis includes different Zeeman splittings at different microwave frequencies, which allows us to disentangle the changes in the g -factor as well as the tip field and reveals changes of around 5% in the g -factor (see Fig. 2(c)). This can account for our smaller than

usual tip fields. Furthermore, the tip fields can vary greatly between different microtips, which is not very controllable due to the random nature of the tip making process. We, therefore, believe that our results present no contradiction with the published literature, e.g. the aforementioned PRL (2017).

c. We do indeed see slight changes in the asymmetry of the predominantly symmetric lineshapes of the ESR peaks at different current setpoints, bias voltages and/or tip-sample distances. Addressing this point in more detail would, however, go beyond the scope of this paper.

5. A general question about the mechanism of the spin-electric coupling: Is it a possibility that not also other degrees of freedom besides the spin are affected by the applied biases and can lead to similar effects? E.g. the crystal field might be affected by the strong local electric fields since the MgO is heavily ionized. This would change the whole energy landscape of the adsorbed molecule. What role does anisotropy play here? And along these lines, is this method even viable for spins $> \frac{1}{2}$? Have the authors tried the same procedure on the Fe atoms that are co-deposited?

Indeed, we believe that the crystal field is affected by these strong local electric fields, which leads to a change in the g -factor as discussed in Ref. [30]. There may certainly be more quantities that are affected by the electric field, which provides exciting perspectives for future research opening up a number of new possibilities for spin manipulation. Anisotropy may also play a role for higher order spins, which likely are just as susceptible to the electric field. This also presents interesting directions for future work. We unfortunately have not been able to test this procedure on Fe atoms due to experimental limitations in our setup.

6. In terms of the TiH molecule, it has been shown that the Oxygen-site molecules exhibit i) an orbital excitation at around 90 meV (ref. [31]) and ii) anharmonic excitations of the dynamic quantum system consisting of hydrogen and Ti atom starting from ca. 35 meV. How can it be excluded that at biases above these thresholds such excitations do not play a role in the interpretation of the data?

The majority of our study and almost all presented data focuses on bridge-site molecules, which exhibit none of the mentioned spectroscopic signatures from orbital excitation as in the oxygen-site TiH nor do they show any signs of anharmonic excitations around 35 meV. This is likely due to changes in the chemical structure and the environment of the TiH, which may be caused by the change in binding site. On the other hand, for measurements on oxygen-site molecules that are presented in the Supplementary Information (Fig. S8), we could not measure above a certain bias voltage (~ 80 meV). This may indeed be due to mentioned orbital excitations coming into play making our system unstable for measurements.

Answers to Referee #2:

This manuscript explores the potential for controlling spin transitions through the application of a constant electric field generated between the tip of an STM and the surface atom. The authors devote half of their research to investigating spin control in an isolated TiH, while the remaining half is focused on the study of two different TiH dimers. The study is well presented and represents a significant contribution to this field. The ability to control spins electrically using these techniques has been a subject of interest for a relatively brief period (last two years may be). This study is the first to clearly demonstrate (experimentally) the feasibility of such control.

Overall, the authors' work is an important contribution to the field and sets a new standard for future research.

We appreciate the referee's assessment of our work.

Despite what was said above, I believe that the authors need to improve the theoretical description of the observations made in this experiment. Specifically, it appears that Equation (1) in the manuscript is incomplete and/or may be lacking some essential information that is required to fully understand its meaning and implications. The total field experienced by the TiH arises from the external magnetic field and the field generated by the tip, as the authors point in this work. However, in the paper by Willke et. al. (*Willke, et al "Magnetic resonance imaging of single atoms on a surface" Nature Physics, 15 (10), (2019)*), it is shown that the field created by the STM tip has two contributions: the dipolar and exchange components. While the present study briefly mention these contributions, they are not discussed in detail. Willke et al. clearly demonstrate that both contributions are substantially different (see Eq about dipolar field and discussion below), particularly regarding the exchange's effect on the resonant frequency, which is independent of g (the effect of the exchange field on the resonance frequency does not depend on g). It is unclear how this is reconciled with equation (1) in the current study. In order to write the resonance frequency as in Eq. (1) we need that the tip's field depends on g . This creates some ambiguity and potentially contaminates the interpretation of Fig. 2 of the present manuscript (c and d). Therefore, the authors should improve the discussion of Eq (1), discussion of Fig. 2, and above all, discussion of the dependence of the field created by the tip on the electric field because, as Eq. (1) is presented, we would have an implicit dependence of the magnetic field on the g -factor.

We thank the referee for their insightful comment but we have to respectfully disagree with the referee's assessment. We do not believe that Eq. 1 in our manuscript contradicts anything presented by Willke *et al.* nor does it imply a dependence of the tip field on the g -factor. In fact, our Eq. 1 is the same equation as Eq. 1 in the paper by Willke *et al.* assuming a spin- $\frac{1}{2}$ system:

$$E_Z = hf_{\text{res}} = g\mu_B B_{\text{ext}} + g\mu_B B_{\text{tip}} = g\mu_B (B_{\text{ext}} + B_{\text{tip}}). \quad (\text{R1})$$

The offset $g\mu_B B_{\text{tip}}$ can be interpreted as an energy offset in the Zeeman splitting or as a shift in the magnetic field by factoring out $g\mu_B$. The energy offset will depend on both the g -factor and the tip field, but this does not imply that the tip field depends on the g -factor. Our analysis actually allows to separate the bias voltage and tip distance dependence of the g -factor and the tip field by extracting the slope and the energy offset/field shift independently, which has not been considered before. In short, the g -factor is related to the slope of the linear fit and the tip field is related to the offset. We agree that B_{tip} has several contributions from dipole and exchange terms, but separating these contributions would require magnetic resonance imaging as in Willke *et al.*, which presents an interesting topic for future work, but does not provide additional insight towards the conclusions in our manuscript. We hope that we could clarify that there is no contradiction to the work by Willke *et al.*

After clarifying the issue with equation (1), I believe it would be interesting to have a more detailed discussion about the different mechanisms involved in the electrical control. It is known that the electric field of the tip modulates both the g -factor and the magnetic field of the tip (as shown in figures 2c and 2d). Have the authors been able to place the STM tip in positions where its field (of the tip) is not present (as shown in the reference [39] of this paper and known as notinfluence point - NOTIN)? This would allow for control to be achieved mainly through the g -factor. It would be helpful if the authors could provide some discussion about this issue.

This is an interesting point. Actually, we believe that we have measured a no tip influence point, which can be seen in Fig. 2(d) of the manuscript. At about -140 meV, we see that the tip field for $I_{\text{sp}} = 250$ pA vanishes. We have added a sentence to the end of paragraph six of our manuscript to highlight this point.

I think that the second half of the manuscript is quite nice. The discussion about control in dimmers is, from my point of view, the most interesting part of the work. The figure 3-d is a good one and, I think it gives us a complete idea of what is possible to do in this kind of experiments.
The conclusions of the work are also clear and insightful, making the reading experience quite enjoyable.
I highly recommend that the authors address the issue highlighted at the beginning of this report. Once this concern has been effectively addressed, I believe that this work should be considered for publication in Nature Communications.

We thank the referee again for the positive assessment and their enthusiasm towards our work.

Answers to Referee #3:

The manuscript describes a detailed experimental investigation of the control of TiH spin transitions indicating that the resonance condition can be tuned by crystal field modulation of the g factor, as well as the STM tip's local field. These dependencies are then leveraged and applications are demonstrated in coupled quantum systems. The paper advances the quantitative understanding of spin systems in STM and importantly the techniques shown as proof of principle here can be used in future investigations of atomic scale quantum systems. The authors provide an excellent level of detail on their experimental setup and methodologies and the quality of the data and presentation are very good. I recommend the paper for publication in Nature Communications after addressing some minor concerns.

We thank the referee for their positive assessment of our work.

1. Abstracts should contain important details such as the experimental system under study (TiH) and key results. The only thing that is quantitative in it is the statement "about ten times its line width". The current abstract is too generic. Control of g and local tip field should be mentioned, as well as applications to tuning interactions of coupled quantum systems. Quantitative results are encouraged.

We have adjusted the second half of our abstract to make it less general adding more specific information while staying within the 150 word limit.

2. "However, so far the bias voltage in ESR-STM has not been employed for spin manipulation" This is confusing because RF modulation of the bias voltage is what drives spin transitions in ESR-STM. I think the authors mean the average (DC) value of the bias voltage here - could they find a way to clarify what they mean?

We thank the referee for pointing out how this sentence is confusing. We have adjusted this sentence in the main text to specify the dc bias voltage.

3. Hopefully I didn't miss it, but please state somewhere in the paper or supplementary if Usp refers to voltage applied to tip or sample for clarity.

We have added two sentences in the Methods section specifying where the bias voltage is applied in our experiments. We thank the referee for this suggestion.

4. I don't agree with the use of "ultrafast" in the abstract and second paragraph (no fs time scales).

We thank the referee for pointing out this oversight. We have changed instances where we write "ultrafast" to "fast" in our manuscript.

5. "The horizontal features in Fig 2(a) and (b)" – have the authors considered these horizontal background features could be due to rectification of the sinusoidal 20mV RF bias through the nonlinear $I(V)$ curve of the junction? That could be checked by repeating the DC bias voltage sweep while superimposing a non-microwave frequency AC modulation (e.g. 100kHz), or by numerically convolving the $I(V)$ curve with an arcsine distribution and subtracting the non-convolved $I(V)$. The authors' explanation is more exotic, but I'd like to make sure there is evidence for it.

We believe that what we wrote in the manuscript and what the referee suggested to be one and the same thing, just argued from a different point of view and, therefore, worded differently. We have adjusted this sentence in the manuscript for clarification.

6. I had difficulty following the fitting procedure on my first readings of the manuscript. I'm pretty sure I understand it now - it's not very complicated, but the sweeps over multiple parameters made it hard to follow what was going on in my first readings. Here are some suggestions to make it easier to follow:

6a. Supp. page 2: "We then use the spline fits at four different microwave frequencies and perform a linear fit at each bias voltage."

I would suggest detailing the procedure a bit more here. I think the spline is mostly a means of tracking the resonance peak position in the U_{bias} , B_{ext} space and to bridge the gap across +/-20mV.

"We then use the spline fits at four different microwave frequencies as a means to query the peak position for an array of bias voltages (including those that pass through 0V). At each bias voltage, the linear Zeeman energy is fit to four points and yield the g-factor..."

I would suggest the authors perhaps add discrete markers on the U_{bias} +/- 150 fits showing the four points the fit line passes through.

6b. I am curious about some of the details of the spline fit. Please consider including some as you see fit. Does the spline fit pass through every maximum peak position in the U_{bias} , B_{ext} heat maps? Does it offer any de-noising or smoothing in the U_{bias} , B_{ext} space (does it have less degrees of freedom than the number of U_{bias} B_{ext} values that were swept?). How were the peak positions determined as input to the spline fit (local fitting? max(signal) point?). It seems like the spline fit is a tool to query the resonance condition at any U_{bias} , enabling a series of fits across U_{bias} and in particular through zero. Is that how I should think of it?

6c. Figure S3 (a) should have a label or caption indicating $f = 61.545$ GHz. Fig S3(b) should have a label or caption indicating $I_{\text{sp}} = 250$ pA.

6d. Supp. page 2: "along with the ESR resonance positions calculated from the fitted values". It's not clear if this refers to the spline fits or the linear fits, recommend to be more explicit: "calculated from the g and B_{tip} fit values for each U_{bias} , I_{sp} condition"

6e. Figure S4: Y axis label should be "External Magnetic Field" rather than "ESR position" – ESR position sounds like it might be the Zeeman energy which it isn't, and the term External Magnetic Field is used consistently elsewhere in the paper.

6f. Supp page 2 "For the frequency dependence of the g-factor" is confusing (makes it sound like g factor is frequency dependent). I think the authors mean "For the frequency shift due to the g-factor bias dependence"

We have changed several parts of the supplementary information to make the spline interpolation and linear fitting more clear.

a) We have changed sentences five, six and nine in paragraph two of section three in the Supplementary Information. We have also changed all instances where we write "spline fits" to "spline interpolations". We hope this will help the reader know more clearly when we are discussing spline interpolations and linear fits. We did not add the discrete markers, because they would not provide any more insight. Instead, we have

made Figs. S4 and S5, which directly show the differences between the extracted peak positions and the peak positions calculated from the linear fits. We have also added a couple of explanatory sentences to help the reader understand the context of these figures.

b) To summarize, the spline interpolation is indeed a way to query the resonance positions in each magnetic field/bias voltage map allowing for us to perform linear fits over a continuous range of bias voltages. The peak positions were found by extracting the peak maxima over a range of bias voltages except within ± 20 mV, where the peak is generally not very well visible. At each bias voltage, we would find the magnetic field value at which the signal is maximal. Then we would apply a spline interpolation over all of these found maxima including the region within ± 20 mV. The spline interpolations include some slight smoothing.

“It seems like the spline fit is a tool to query the resonance condition at any U_{bias} , enabling a series of fits across U_{bias} and in particular through zero. Is that how I should think of it?” – Yes, that is the idea.

c) We have added the radio frequency and current set point of the measurements presented in the Fig. S3 in the caption..

d) We have adjusted the first and second sentence of paragraph three in section three of the supplementary.

e) We have changed the y -axis label for Fig. S4 to “External Magnetic Field”.

f) We have changed “For the frequency dependence of the g -factor” to “For the frequency shift due to the g -factor bias voltage dependence”.

We thank the referee for these suggestions.

Reviewers' Comments:

Reviewer #1:

Remarks to the Author:

I thank the authors for sending in their improved version of the manuscript. I only have a few follow-up remarks left on this, through which I will go point by point. If these are addressed, I would recommend the publication of this article.

1. Ok, that's better. Small typo in one added sentence now: "Similarly to what has be shown previously". Be should be been.
2. Interesting point, thanks for clarification.
3. Ok, I think this is much more clear now.
4. Overall, this point is much clearer to me now. But I would like to ask the authors to add a sentence or two about their clarification of point 4b) into the manuscript as well. I believe this is an important point that might not be vastly obvious to the general reader.
5. I am missing here a discussion about this in the paper. I would like to ask the authors (in their conclusions or in their discussion about the SEC) to give a little bit of a critical discussion of this method as well. What are possible hurdles or where does one have to be particularly careful. As the authors say this is not diminishing the work but rather showing there is potential for this method for future experiments.
6. I thank the authors for clarification. Again I believe that this is interesting information for the community and would like to ask the authors to add a sentence about the latter part of their reply into the supplemental part of the manuscript.

Reviewer #2:

Remarks to the Author:

I appreciate the dedication of the authors to answer my questions and dispel most of my doubts.

The employed method for obtaining "g" is reliable. By keeping the tip at a fixed distance (maintaining a constant value of "g * B_{tip}") it becomes feasible to determine "g" through the process of sweeping the External Magnetic Field and using equation R1 (Eq 1 of the preprint) for fitting. Figure S3 is a good one to see this.

Despite this, I still think that Eq. 1 of the work, in its current form and without any accompanying explanation, can be perceived as somewhat confusing. Indeed, it is accurate to say that Eq. 1 in the present work seems to be identical to Eq. 1 of the paper conducted by Willke et.al. However, it is very important to note that the latter work provides two significant clarifications about the tip field. Firstly, they present Eq. 2, which defines the dipolar tip field. Secondly, in the subsequent paragraph, they define the effective tip field related to the exchange interaction, where the field is dependent on "g". Eq. 2 shows that the effective dipolar field does not depend on the g-factor. Clearly this makes it difficult to extract "g" as a common factor in a clean and clear way. A similar observation can be made by examining Eq. 4 in the paper conducted by Rodriguez et.al (Physical Review B vol 107, 155406 (2023)), which further reinforces this point.

As illustrations, let us examine two scenarios. Firstly, when the tip is positioned far away from the adatom. Here, the interaction between the tip and the atom can be characterized as only dipolar. In this particular case, Eq. 1 from the submitted article is, from my point of view, unequivocally accurate. Conversely, when the tip is very close to the atom, the interaction can be assumed as purely exchange. In this case the term "g μ_B B_{tip}" of Eq. 1 becomes independent of the factor "g." Intermediate distances between the tip and atom present a more intricate situation, where the analysis becomes more complex.

Based on the information provided, I'm sure that the work does not have significant issues regarding the accurate estimation of "g" and the corresponding analyses associated with this parameter. However, in light of the mentioned concerns about the tip field and its dependence on the "g" factor, it would be beneficial to consider incorporating additional clarifications after Eq. 1. Moreover, it is crucial to thoroughly review the conclusions and the figures related to the behavior of "Btip". This holds significant importance to me, due to the observations presented in figure 2d, showcasing intriguing behaviors of Btip.

While I understand that there exist the possibility of error in my reasoning, it is clear that discrepancies exist between the statements made in Eq. 1 of this particular work and the equations presented by Willke (his Eq. 1, Eq. 2 and explanations bellow) and Rodriguez (Eqs. 1 to 9). In the event that I have overlooked certain aspects that could influence my analysis, I kindly request the authors to provide a more detailed explanation than what was provided in the initial response clarifying why I'm wrong.

My position remains the same. After clarifying this particular point, I believe that the work under consideration deserve publication Nature Communications.

Reviewer #3:

Remarks to the Author:

The authors have answered the concerns in my initial review. I recommend the manuscript for publication in Nature Communications.

Second Reply to Referee's Comments

Changes in the manuscript are marked in blue.

Answers to Referee #1:

I thank the authors for sending in their improved version of the manuscript. I only have a few follow-up remarks left on this, through which I will go point by point. If these are addressed, I would recommend the publication of this article..

We thank the referee for their positive review of our work.

1. Ok, that's better. Small typo in one added sentence now: "Similarly to what has been shown previously". Be should be been.

We thank the referee for pointing this typo out and we have corrected it.

2. Interesting point, thanks for clarification.

3. Ok, I think this is much more clear now.

4. Overall, this point is much clearer to me now. But I would like to ask the authors to add a sentence or two about their clarification of point 4b) into the manuscript as well. I believe this is an important point that might not be vastly obvious to the general reader.

We thank the referee for this suggestion and we have added two sentences to the end of paragraph 6 of our main text.

5. I am missing here a discussion about this in the paper. I would like to ask the authors (in their conclusions or in their discussion about the SEC) to give a little bit of a critical discussion of this method as well. What are possible hurdles or where does one have to be particularly careful. As the authors say this is not diminishing the work but rather showing there is potential for this method for future experiments.

We have added several sentences into the conclusion based on this comment. We thank the referee for their suggestion.

6. I thank the authors for clarification. Again I believe that this is interesting information for the community and would like to ask the authors to add a sentence about the latter part of their reply into the supplemental part of the manuscript.

We thank the referee for this comment and have added a sentence to the end of section VI in our supplementary.

Answers to Referee #2:

I appreciate the dedication of the authors to answer my questions and dispel most of my doubts.

The employed method for obtaining "g" is reliable. By keeping the tip at a fixed distance (maintaining a constant value of " $g \cdot B_{\text{tip}}$ ") it becomes feasible to determine "g" through the process of sweeping the External Magnetic Field and using equation R1 (Eq 1 of the preprint) for fitting. Figure S3 is a good one to see this.

We greatly appreciate the Referee's discussion of this point.

Despite this, I still think that Eq. 1 of the work, in its current form and without any accompanying explanation, can be perceived as somewhat confusing. Indeed, it is accurate to say that Eq. 1 in the present work seems to be identical to Eq. 1 of the paper conducted by Willke et.al. However, it is very important to note that the latter work provides two significant clarifications about the tip field. Firstly, they present Eq. 2, which defines the dipolar tip field. Secondly, in the subsequent paragraph, they define the effective tip field related to the exchange interaction, where the field is dependent on "g". Eq. 2 shows that the effective dipolar field does not depend on the g-factor. Clearly this makes it difficult to extract "g" as a common factor in a clean and clear way. A similar observation can be made by examining Eq. 4 in the paper conducted by Rodriguez et.al (Physical Review B vol 107, 155406 (2023)), which further reinforces this point.

We agree with the Referee on this point. Depending on the contributions from a dipolar tip field and the exchange interaction, the tip field may effectively be more or less dependent on the g-factor. However, since the g-factor only varies by about 1% in the whole range of Fig. 2(c), the effect on the tip field in Fig. 2(d) will be very small.

As illustrations, let us examine two scenarios. Firstly, when the tip is positioned far away from the adatom. Here, the interaction between the tip and the atom can be characterized as only dipolar. In this particular case, Eq. 1 from the submitted article is, from my point of view, unequivocally accurate. Conversely, when the tip is very close to the atom, the interaction can be assumed as purely exchange. In this case the term " $g \mu_B B_{\text{tip}}$ " of Eq. 1 becomes independent of the factor "g." Intermediate distances between the tip and atom present a more intricate situation, where the analysis becomes more complex.

In principle, we again agree with the Referee on this point. However, as we determine the tip field itself, its dependence on the g -factor would only become relevant, if we wanted to disentangle the contributions of the dipolar interaction and the exchange interaction. Still, the extraction of the g -factor from the slope is accurate and reliable (as confirmed by the Referee), since the slope only depends on the g -factor, so that the extracted g -factor could be used to further disentangle the different contributions to the tip field.

Based on the information provided, I'm sure that the work does not have significant issues regarding the accurate estimation of "g" and the corresponding analyses associated with this parameter. However, in light of the mentioned concerns about the tip field and its dependence on the "g" factor, it would be beneficial to consider incorporating additional clarifications after Eq. 1. Moreover, it is crucial to thoroughly review the conclusions and the figures related to the behavior of "Btip". This holds significant importance to me, due to the observations presented in figure 2d, showcasing intriguing behaviors of Btip.

We want to reiterate that the slope of the first order polynomial fit function only depends on the g -factor, so that we believe analysis of the g -factor to be accurate. We also believe the analysis of the tip field itself to be accurate. However, we can only speculate about the different contributions to the tip field. Separating these contributions including their dependence on not just the g -factor, but also the bias voltage and the tip-sample distance, would go beyond the scope of this paper. We have included a statement after Eq. (1) that clarifies these points and explains the g -factor dependence of the tip field. Finally, we want to point out that, as mentioned above, the g -factor only changes by about 1% in the measured parameter space and the tip field increases with decreasing g -factor, both of which cannot explain the intriguing behavior of the tip field.

While I understand that there exist the possibility of error in my reasoning, it is clear that discrepancies exist between the statements made in Eq. 1 of this particular work and the equations presented by Willke (his Eq. 1, Eq. 2 and explanations below) and Rodriguez (Eqs. 1 to 9). In the event that I have overlooked certain aspects that could influence my analysis, I kindly request the authors to provide a more detailed explanation than what was provided in the initial response clarifying why I'm wrong.

My position remains the same. After clarifying this particular point, I believe that the work under consideration deserve publication Nature Communications.

We appreciate the referee's interesting discussion and we hope that we could clarify their concerns, so that our manuscript is ready for publication in Nature Communications.

Answers to Referee #3:

The authors have answered the concerns in my initial review. I recommend the manuscript for publication in Nature Communications.

We thank the referee for their recommendation to publish our work in Nature Communications.